# Electrospinning Composites as Carriers of Natural Pigment: Screening of Polymeric Blends

**Sergiana dos Passos Ramos** [1], **Letícia Guerreiro da Trindade** [2] , **Tatiana Martelli Mazzo** [3], **Elson Longo** [4],
**Fabiana Perrechil Bonsanto** [2] , **Veridiana Vera de Rosso** [5] and **Anna Rafaela Cavalcante Braga** [1,2,*]

[1] Department of Biosciences, Universidade Federal de São Paulo (UNIFESP), Santos 11015-020, SP, Brazil
[2] Department of Chemical Engineering, Universidade Federal de São Paulo (UNIFESP), Diadema 09972-270, SP, Brazil
[3] Institute of Marine Sciences, Universidade Federal de São Paulo (UNIFESP), Santos 11015-020, SP, Brazil
[4] Chemistry Department, CDMF/LIEC (UFSCar), São Carlos 13560-970, SP, Brazil
[5] Nutrition and Food Service Research Center, Universidade Federal de São Paulo (UNIFESP), Santos 11015-020, SP, Brazil
* Correspondence: anna.braga@unifesp.br

**Abstract:** Several studies have already demonstrated that electrospinning is an excellent tool for forming nano/microfibers. However, the number of parameters affecting the formation of the structures has become a great challenge, including the polymeric solutions' rheological properties, directly affecting the morphology of the fibers formed. The present work aimed to produce polymeric composites and determine their rheological properties, comparing them to the morphology of the fibers formed by electrospinning. Also, to evaluate their potential use as the carriers of natural pigments. To this end, a distinct combination of solutions containing Chitosan/Gelatin, Chitosan/poly(ethylene) oxide (PEO) and Zein/PEO was produced and submitted to electrospinning. The sample containing zein manufactured the structures smaller in diameter (201.3 ± 58.6 nm) among those studied. Besides, it was observed that adding PEO to the solutions impacts the increase in viscosity and shear thinning behavior, guaranteeing uniformity in the structures formed. Natural pigments were successfully incorporated into the chosen zein/PEO solution, and it was observed that adding these compounds led to changes in the rheological characteristics, as expected. Nevertheless, it was possible to produce uniform fibers with diameters ranging from 665.68 ± 249.56 to 2874.44 ± 1187.40 nm, opening the possibility of using these natural pigments in biotechnological processes.

**Keywords:** electrospun; composites; zein; PEO; *Spirulina*; C-phycocyanin; carotenoids; anthocyanins; rheological properties

## 1. Introduction

Incorporating natural pigments into biotechnological products to improve their quality is highly interesting to the food, cosmetics and pharmaceutical industries [1–3]. Antioxidants, antimicrobials and pigment compounds are typically present in food and cosmetic products in small quantities but are frequently exceptionally active [4]. On the other hand, as vastly reported by the literature, they are known as unstable regarding the application of industrial processes involving high temperatures or pH variation [5,6].

Encapsulation is a powerful tool to overcome many of the previously mentioned drawbacks because it protects a great variety of compounds by covering the active molecules with a protective matrix [7–9], and electrospinning has shown great potential to produce electrospun fibers to overcome the challenges of applying natural pigments.

Different methods can be applied to aid the protection of the biomolecules that present biological effects, particularly to produce nanostructures. The most reposted include the manufacture of nanoparticles, nanoemulsions, nanodroplets and nanohydrogels. The

methodologies of choice rely on varied parameters covering chemical characteristics, physical properties, particle size and the release ratio of the target compound [10,11].

Electrospinning is a simple method to fabricate composite, mostly fibers, from micro- to nanometer diameter throughout an electrically charged jet of polymeric solutions. Electrospinning fibers present exceptional properties such as a high surface area to volume ratio, one of the most highlighted features in producing composites. Furthermore, mild temperature conditions are required in the process, which is excellent for bioactive compound encapsulation. Electrospinning permits the utilization of a wide variety of food-grade, biodegradable, biocompatible polymeric substances as wall materials for encapsulating bioactive compounds. Among the food-grade polymer options, PEO, zein, gelatin and chitosan present several reported features to be used in bioactive compound encapsulations [12–14].

Polyethylene oxide has been widely used as a polymeric matrix for incorporating bioactive compounds due to its classification as GRAS (Generally Recognized as Safe), water solubility and ability to confer elasticity to the formation of fibers obtained by electrospinning, and the consequent reduction of diameters [11]. However, the decline in the use of synthetic polymers and their partial replacement by natural polymers have been highlighted by blends of synthetic polymers and natural polymers such as proteins, lipids and polysaccharides [10]. Zein is the main reserve protein of corn and is one of the most hydrophobic proteins due to the large presence of apolar amino acids in its structure. In addition to biodegradability, zein is thermally stable and can form uniform films [14]. Chitosan is an amino polysaccharide obtained from chitin. It has good film forming ability and antibacterial properties [15]. Gelatin is a mixture of high molecular weight peptides derived from collagen, abundant in skin, bone and cartilage. This peptide combined with chitosan can lead to the formation of stable and flexible films [16].

In this sense, fluids' rheological properties are critical to process optimization and understanding the material behavior, mainly when using electrospinning. However, a few studies have been conducted to correlate the rheological properties of fiber-forming solutions with fiber formation and morphology [17] without reaching a consensus.

The present work aimed to produce composites as the carriers of natural pigment using electrospinning and to conduct a screening of polymeric blends, determining the polymeric solutions' rheological properties and attempting to correlate the rheological properties of polymeric solution with the microscopic characteristics of electrospun fibers.

## 2. Materials and Methods

### 2.1. Solutions Preparation for Electrospinning

A search was performed in the literature to execute the screening of polymeric blends, and the chosen references are presented in Table 1. The solutions were prepared according to the author's recommendations.

### 2.2. Natural Pigment Incorporation

A search was performed in the literature to execute the screening of polymeric blends, and the chosen references are presented in Table 1. The solutions were prepared according to the author's recommendations. To determine the incorporation behavior of a range of bioactive compounds with different chemical properties, *Spirulina* biomass, C-phycocyanin (C-PC), carotenoids and anthocyanins were obtained and incorporated into the electrospun polymeric fibers.

**Table 1.** Parameters (solution composition, applied voltage, feeding rate, tip-to-collector distance (TCD)) based on the literature used to evaluate the production of electrospinning nanostructures.

| Sample | Composition (%) | Voltage (kV) | Feeding Rate (μL/h) | TCD (cm) | Reference |
|--------|-----------------|--------------|---------------------|----------|-----------|
| 1 | Chitosan/gelatin (25/3) | 24 | 800 | 15 | [18] |
| 2 | Chitosan/PEO (2/3) | 20 | 300 | 10 | [15] |
| 3 | Zein (20) | 20 | 1000 | 20 | [19] |
| 4 | Gelatin (10) | 25 | 1800 | 10 | [20] |
| 5 | PEO/zein (12/0.3) | 20 | 1800 | 15 | [21] |
| 6 | PEO/zein (12/1) | 20 | 1800 | 15 | [21] |
| 7 | PEO/gelatin (10/2) | 25 | 2000 | 13 | [20] |
| **Sample** | **Composition (%)** | **Voltage (kV)** | **Feeding Rate (μL/h)** | **TCD (cm)** | **Pigment** |
| 8 | PEO/zein (12/1) | 20 | 1800 | 15 | *Spirulina* |
| 9 | PEO/zein (12/1) | 20 | 1800 | 15 | C-phycocyanin |
| 10 | PEO/zein (12/1) | 20 | 1800 | 15 | Carotenoid (from pitanga) |
| 11 | PEO/zein (12/1) | 20 | 1800 | 15 | Anthocyanin (from jussara) |

*Spirulina* biomass was kindly donated by Fazenda Tamanduá® (Santa Teresinha, Brazil). According to Fatrelli et al. [22], the methodology developed by our research group, the C-PC, was extracted from *Spirulina*. Carotenoids were extracted from pitanga, following the De Rosso et al. [23] method. Anthocyanin extract was obtained from jussara pulp, according to Giaconia et al. [24]. The conditions used were the same as sample 6, adding the mentioned natural pigments (polymeric solution containing 1% of PEO and 12% of zein).

To prepare the solution containing *Spirulina* biomass, 12% (*w/v*) zein (Sigma Aldrich, St. Louis, MO, USA) and 1% (*w/v*) PEO (900,000 g.moL$^{-1}$, Sigma Aldrich, St. Louis, MO, USA) was dissolved in ethanol 80%. *Spirulina* was added into the zein/PEO solution at a concentration of 20% (*w/v*) and then mixed at room temperature under constant stirring until complete homogenization. The same procedure was applied to prepare C-PC polymeric solution. After zein/PEO mixing, C-PC extract was added into the zein/PEO solution at a concentration of 20% (*v/v*) and then mixed at room temperature under constant stirring until complete homogenization.

The carotenoid extract was resuspended in 80% ethanol with a concentration of 400 μg/mL. Zein (Sigma Aldrich, St. Louis, MO, USA) 12% (*w/v*) was dissolved in the carotenoid solution until complete dissolution, and 1% (*w/v*) of PEO (900,000 g.moL$^{-1}$, Sigma Aldrich, St. Louis, MO, USA) was added to the system and then homogenized at room temperature under constant stirring.

Finally, for anthocyanin solutions, zein (Sigma Aldrich, St. Louis, MO, USA) 12% (*w/v*) was dissolved in jussara pulp and reconstituted in a sodium acetate buffer (pH 4.5) until complete dissolution. After that, 1% (*w/v*) of PEO (900,000 g.moL$^{-1}$, Sigma Aldrich, St. Louis, MO, USA) was added to the solution and mixed at room temperature until complete homogenization.

### 2.3. Rheological Analysis of Solutions

The rheological properties of polymeric blends and solution containing the bioactive compounds were evaluated using an MCR 92 rheometer (Anton Paar, Austria) equipped with a parallel plate geometry with a diameter of 25 mm or 50 mm (depending on the viscosity of the solution) and a gap of 1 mm. Flow curves were obtained by an up-down-up step program between 0 and 300 s$^{-1}$. The curves of shear stress versus shear rate were then fitted to the power law model (Equation (1)).

$$\sigma = k \cdot \dot{\gamma}^{n} \tag{1}$$

where $\sigma$ is the shear stress (Pa), $\dot{\gamma}$ is the shear rate (s$^{-1}$), $n$ is the flow behavior index and $k$ is the consistency index (Pa.s$^n$).

The shear rate of the electrospinning process was estimated according to Equation (2), described by Balik et al. [17].

$$\dot{\gamma} = \left(\frac{3n+1}{4n}\right)\frac{4Q}{\pi R^3} \tag{2}$$

where $Q$ is the volumetric flow rate (m$^3$/s) and $R$ is the inside radius of the needle (m).

Frequency sweep tests were performed from 0.1 to 10 Hz within the linear viscoelastic range (maximum deformation of 1%). The parameters determined were the dynamic storage (G'), loss (G'') and tan δ (=G''/G'). All the rheological measurements were performed in triplicate at 25 °C.

### 2.4. Electrospinning Process

The process followed the parameters previously developed and presented in the consulted literature. Briefly, the polymeric structures were prepared using laboratory-scale electrospinning (FLUIDNATEK LE-10, BIOINICIA, Spain) with a steel needle of 0.6 mm diameter [11]. The samples were produced and collected at controlled room temperature (20 to 25 °C) and relative humidity (50% to 60%) [25]. The process and solution parameters for the electrospinning (i.e., the feeding rate, the polymers concentrations, the tip-to-collector distance (TCD) and the applied voltage) were chosen from the best condition parameters from the previous study for nanofiber production and are described in Table 1. The moisture of the electrospun samples was accepted to be removed entirely during the electrospinning process, so the results of all analyses were expressed on a dry weight basis.

### 2.5. Characterization of Electrospun Samples

After the composites were produced, the material was characterized [25]. Field emission scanning electron microscopy (FE-SEM Supra 35 VP- equipment, Carl Zeiss, Germany) was used to obtain the micrographic images of the samples, and the tool DiameterJ (ImageJ program) was applied. Additionally, fundamental vibrational modes and wavenumbers from experimental spectra were obtained using Fourier-transform infrared spectroscopy (FTIR) (Bruker Alpha-P, 4000–500 cm$^{-1}$). Thermal stability of the composites was characterized by thermogravimetric analysis using a TA Instruments Q-50 apparatus (Mettler-Toledo, Barueri—SP, Brazil) under a temperature range of 0–700 °C and an N$_2$ atmosphere with a scan rate of 10 °C.min$^{-1}$ [11,25,26].

### 2.6. Statistical Analysis

All analyses were performed in triplicate and expressed in terms of mean and ±standard deviation. Besides, the measurements from the samples, using the MEV images, were carried out independently. In both cases, samples were compared by applying analysis of variance (ANOVA) using the degree of significance of 95% ($p < 0.05$), followed by Tukey's (more than two samples) post hoc test or test T (comparing two samples).

### 3. Results

#### 3.1. Rheological Analysis of Solutions

The flow curves of the polymeric solutions are shown in Figure 1A, and the rheological parameters obtained from these curves are reported in Table 2. The addition of bioactive compounds affected the rheological behavior of the polymeric solution, as can be seen in Figure 1B and Table 2.

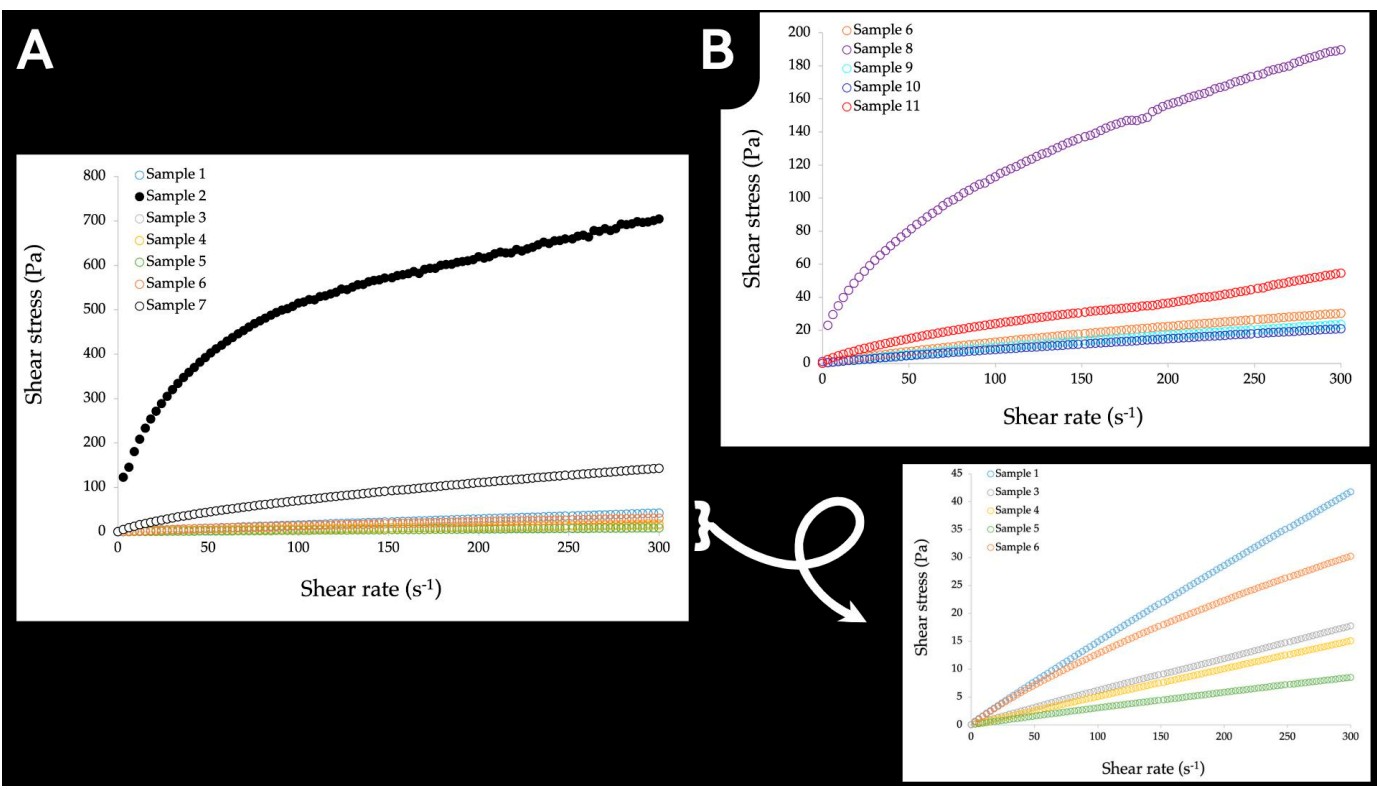

**Figure 1.** Flow curves of (**A**) polymeric solutions and (**B**) solutions containing the bioactive compounds.

The oscillatory behavior of the solutions is shown in Figure 2. For samples that showed a flow characteristic close to Newtonian behavior (samples 1, 3, 4 and 5), the oscillatory measurements could not be performed because these solutions do not have a viscoelastic characteristic. The values of tan δ were higher than 1 for the evaluated solutions (Figure 2A), except for sample 6, which showed tan δ around 1. The addition of *Spirulina* and anthocyanin (samples 8 and 11, respectively) led to a decrease in tan δ for values lower than 1, indicating that these samples began to behave like a solid-like structure (G′ > G″).

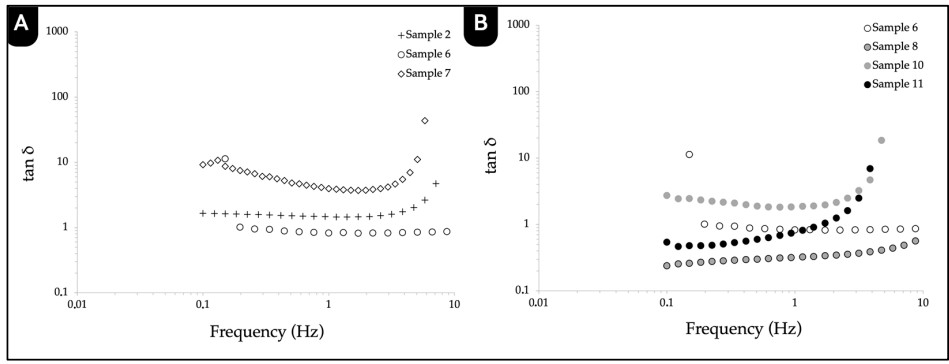

**Figure 2.** Tan δ obtained from frequency sweeps of (**A**) polymeric solutions and (**B**) solutions containing the bioactive compounds.

**Table 2.** Rheological parameters of polymeric blends and solutions containing the bioactive compounds and the diameter of electrospun fibers.

| Sample | Composition (%) | $\dot{\gamma}$ (s$^{-1}$) | n (-) | k (Pa.s$^n$) | η ($\dot{\gamma}$) (mPa.s) | Diameter (nm) |
|---|---|---|---|---|---|---|
| 1 | Chitosan/gelatin (25/3) | 9.4 | 0.94 [d] ± 0.00 | 0.19 [a] ± 0.01 | 172.90 [a] ± 6.71 | - |
| 2 | Chitosan/PEO (2/3) | 0.6 | 0.31 [a] ± 0.03 | 106.94 [b] ± 10.03 | 40,230.33 [c] ± 249.88 | 314.9 [a] ± 427.7 |
| 3 | Zein (20) | 12.3 | 0.96 [d] ± 0.00 | 0.07 [a] ± 0.00 | 63.79 [a] ± 0.34 | 266.8 [a] ± 221.9 |
| 4 | Gelatin (10) | 23.6 | 1.00 [e] ± 0.00 | - | 50.39 [a] ± 0.25 | - |
| 5 | Zein/PEO (12/0.3) | 20.9 | 0.93 [d] ± 0.01 | 0.04 [a] ± 0.00 | 32.89 [a] ± 1.24 | 712.4 [b] ± 415.1 |
| 6 | Zein/PEO (12/1) | 16.2 | 0.81 [c] ± 0.01 | 0.30 [a] ± 0.02 | 153.03 [a] ± 6.73 | 201.3 [a] ± 58.6 |
| 7 | PEO/gelatin (10/2) | 12.7 | 0.65 [b] ± 0.01 | 3.54 [a] ± 0.61 | 1325.83 [b] ± 188.53 | 648.3 [b] ± 260.7 |
| **Sample** | **Composition (%)** | | | | | **Diameter (nm)** |
| 8 | Zein/PEO (12/1)—*Spirulina* | 7.0 | 0.48 ± 0.01 | 12.11 ± 1.02 | 4729.87 ± 340.15 | 702.63 ± 253.08 |
| 9 | Zein/PEO (12/1)—C-phycocyanin | 16.4 | 0.81 ± 0.00 | 0.23 ± 0.00 | 114.78 ± 2.13 | 665.68 ± 249.56 |
| 10 | Zein/PEO (12/1)—Carotenoid | 17.7 | 0.85 ± 0.00 | 0.17 ± 0.00 | 97.82 ± 2.21 | 2874.44 ± 1187.40 |
| 11 | Zein/PEO (12/1)—Anthocyanin | 12.1 | 0.71 ± 0.05 | 0.87 ± 0.31 | 427.77 ± 107.96 | 975.97 ± 330.68 |

Different letters indicate a significant difference ($p < 0.05$) in the same column. $\dot{\gamma}$: shear rate (s$^{-1}$), *n*: flow behavior index, *k*: consistency index and η ($\dot{\gamma}$): apparent viscosity at the estimated shear rate.

### 3.2. Characterization of Electrospun Samples

As already expected, and reported in the literature, variations in the polymeric solution composition can significantly affect the film/fiber diameter and morphology [11,17,20,27]. The present work evaluated the nanofibers' morphologies and average polymeric structure diameters using FE-SEM (Figure 3). The FE-SEM images of samples 1, 2, 3, 4, 5, 6 and 7 can be seen in Figure 3a–n. It is possible to observe changes in the structure of the samples as the composition changes. For samples 1 and 4 (Figure 3a,b,g,h), we observed no fibers forming when adding chitosan. For sample 2 (Figure 3c,d), there is the formation of fibers but with an agglomerated appearance and no shape homogeneity. The pure zein sample 3 (Figure 3e,f) presents the formation of fibers with better shape uniformity. Samples 5, 6 and 7 (Figure 3i–n) present the appearance of fibers with the best shape and diameter definition. Sample 6 can be highlighted as the best because it shows the formation of homogeneous nanofibers in size and orientation. Sample 6 demonstrates a better definition of shape and diameter, which can be attributed to the greater amount of PEO (1%) concerning sample 5 (0.3%).

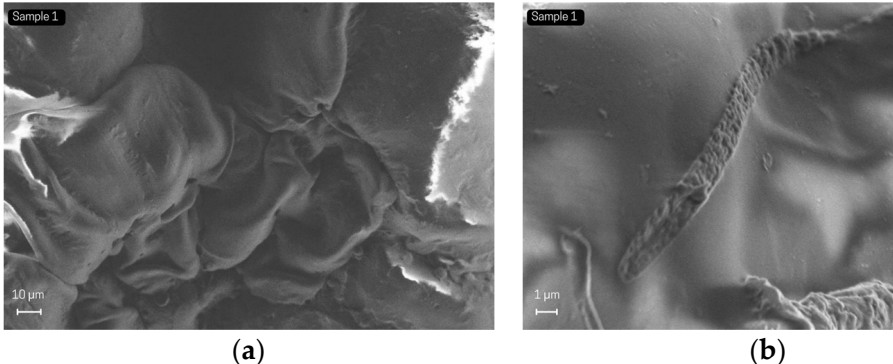

(a)　　　　　　　　　　　　　　　(b)

**Figure 3.** *Cont.*

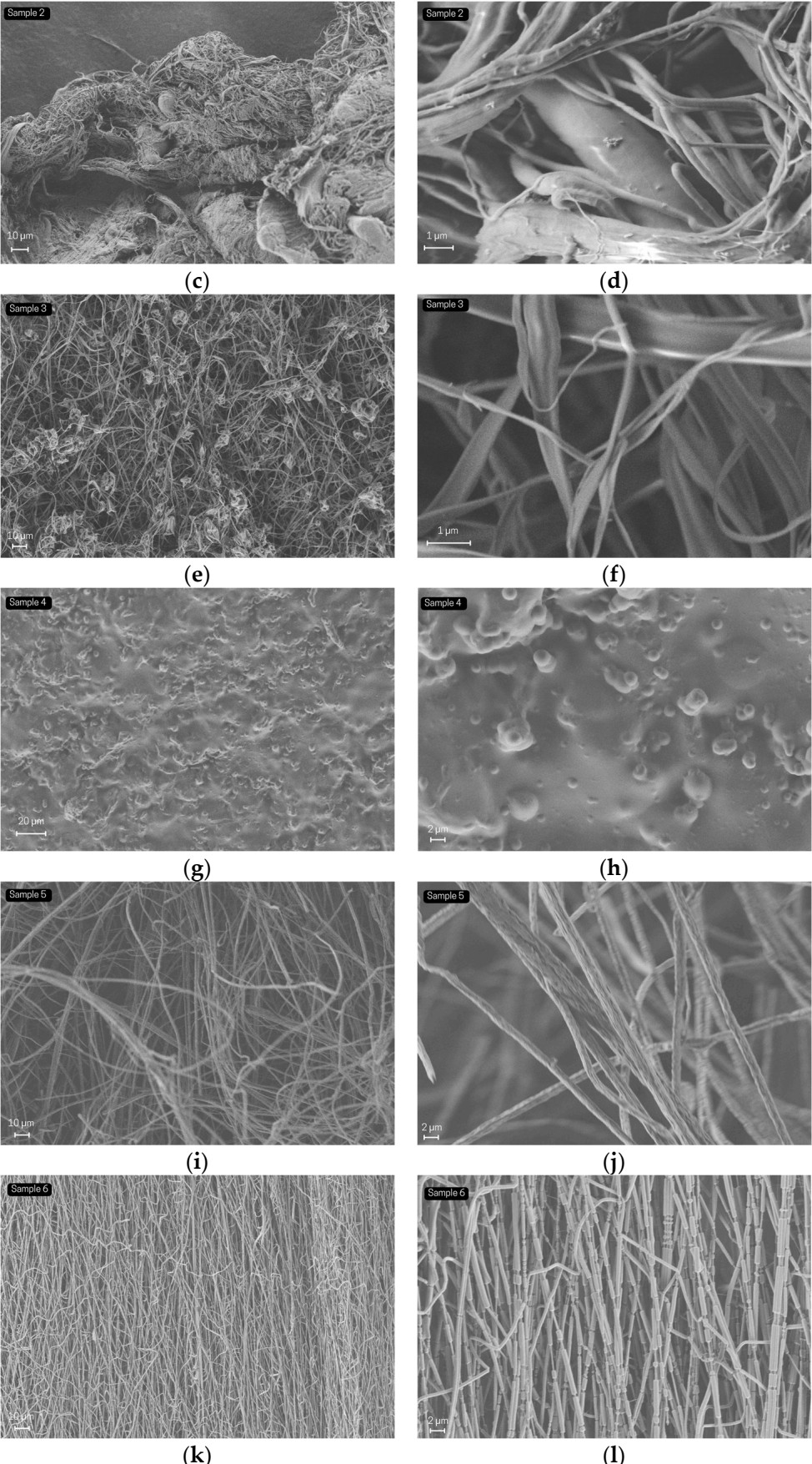

**Figure 3.** *Cont.*

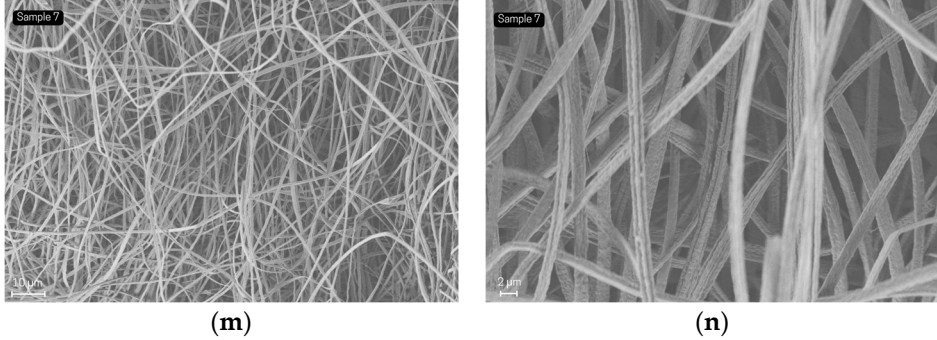

**Figure 3.** Field emission scanning microscopy images of samples 1 (**a,b**), 2 (**c,d**), 3 (**e,f**), 4 (**g,h**), 5 (**i,j**), 6 (**k,l**), 7 (**m,n**).

Figure 4 shows the ATR-FTIR spectra of the samples. To better compare the results, samples 1, 2, 4 and 7 are presented in Figure 4a, and 3, 5 and 6 are displayed in Figure 4b. The analytical evaluation of these samples' spectrum in terms of functional groups assigned for the absorption of certain wavenumbers is given in Table 3.

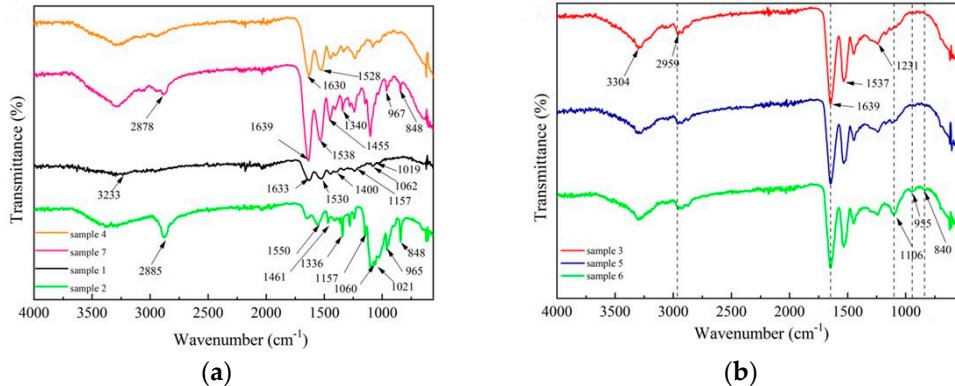

**Figure 4.** ATR-FTIR of samples 1 (**a**), 2 (**a**), 3 (**b**), 4 (**a**), 5 (**b**), 6 (**b**) and 7 (**a**).

**Table 3.** Characteristic absorption peaks assigned from ATR-FTIR spectra for samples 1, 2, 3, 4, 6, and 7.

| Sample | Peak Position on ATR-FTIR Spectra (cm$^{-1}$) | Functional Group Assigned |
|:------:|:---:|:---:|
| 1 | 1530 | C-O-C antisymmetric stretching |
| | 1157 | C-O stretching vibration |
| | 1062 | C-O-C antisymmetric stretching |
| | 1019 | O-H bending vibration |
| | 1400 | symmetric stretching of -COO groups |
| | 1633 | C=O stretching vibration |
| 2 | 2855 | CH$_2$ stretching |
| | 1461 | asymmetric CH$_2$ bending |
| | 1336 | symmetric CH$_2$ wagging and some C-C stretching |
| | 965 | C-O stretching vibration |
| | 848 | antisymmetric stretching vibration of the CH$_2$ group |
| 3 | 3304, 2959 | R-C=O-R' (carbonyl groups) |
| | 1639 | -HN-C=O-R- (amide I) |
| | 1537 | angular deformation vibrations of the N-H bond (amide II) |
| | 1231 | axial deformation vibrations of the C-N bond |
| 4 | 1630 | C=O stretching vibration |
| 6 | 1106 | C-O-C stretching |
| | 955 | CH$_2$-CH$_2$ rocking and C-O-C vibration modes |
| | 840 | CH$_2$ rocking |

**Table 3.** *Cont.*

| Sample | Peak Position on ATR-FTIR Spectra (cm$^{-1}$) | Functional Group Assigned |
|---|---|---|
| 7 | 2878 | C-H stretching vibrations |
| | 1455, 1340 | C-H bending vibrations |
| | 967 | C-O stretching vibration |
| | 848 | antisymmetric stretching vibration of the CH$_2$ group |

Based on the FE-SEM and ATR-FTIR analyses that showed that the combination of PEO and zein polymers (1% and 12%) presented thin tubular fibers with better shapes and diameter, sample 6 was chosen to incorporate the natural pigments. The FE-SEM of the pigment's composites is shown in Figure 5: Spirulina (Figure 5a,b), C-phycocyanin (Figure 5c,d), carotenoids (Figure 5e,f) and jussara anthocyanins (Figure 5g,h).

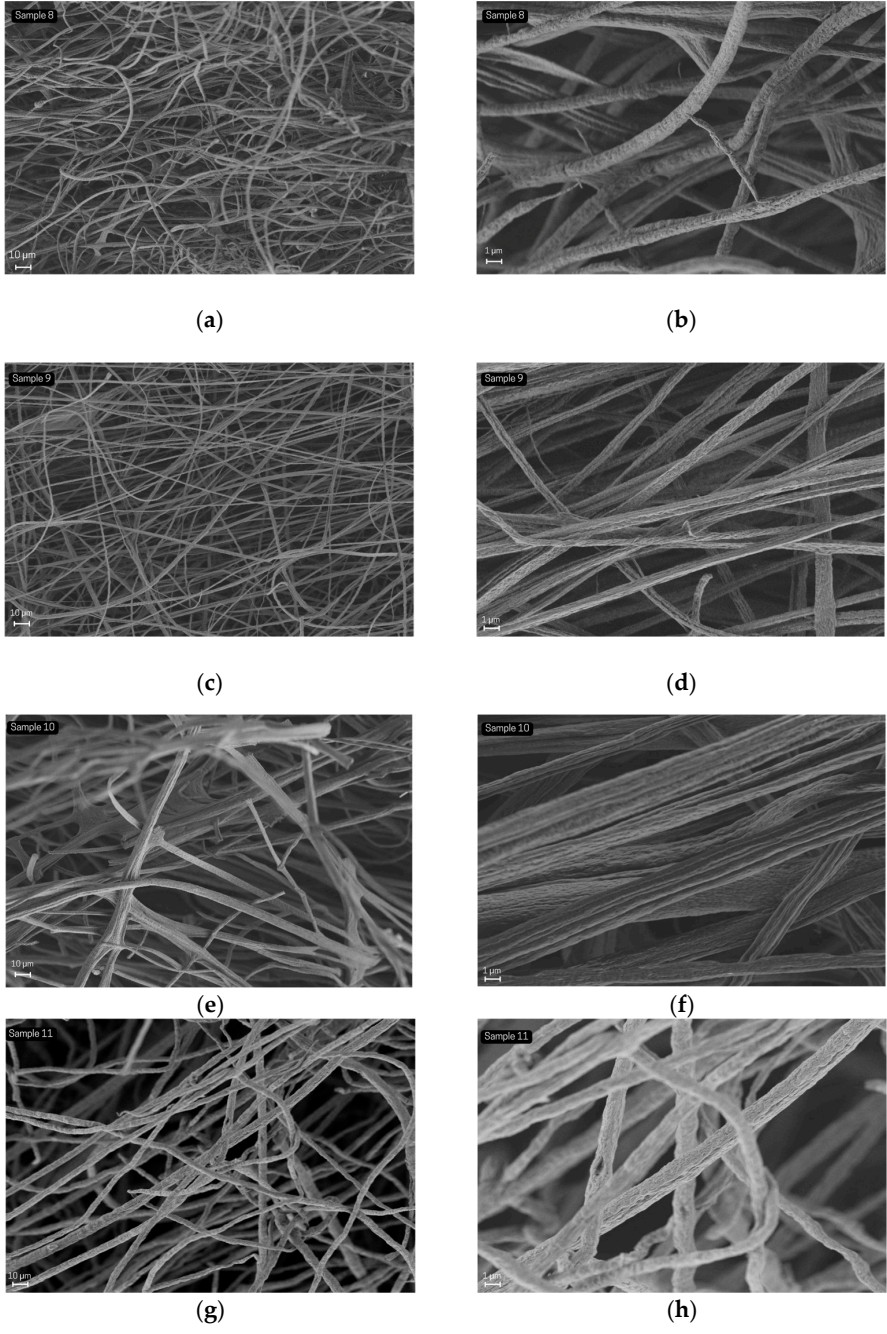

**Figure 5.** Field emission scanning microscopy images of samples 8 (**a,b**), 9 (**c,d**), 10 (**e,f**) and 11 (**g,h**).

The ATR-FTIR spectra of the PEO and Zein polymers modified with natural pigments are shown in Figure 6.

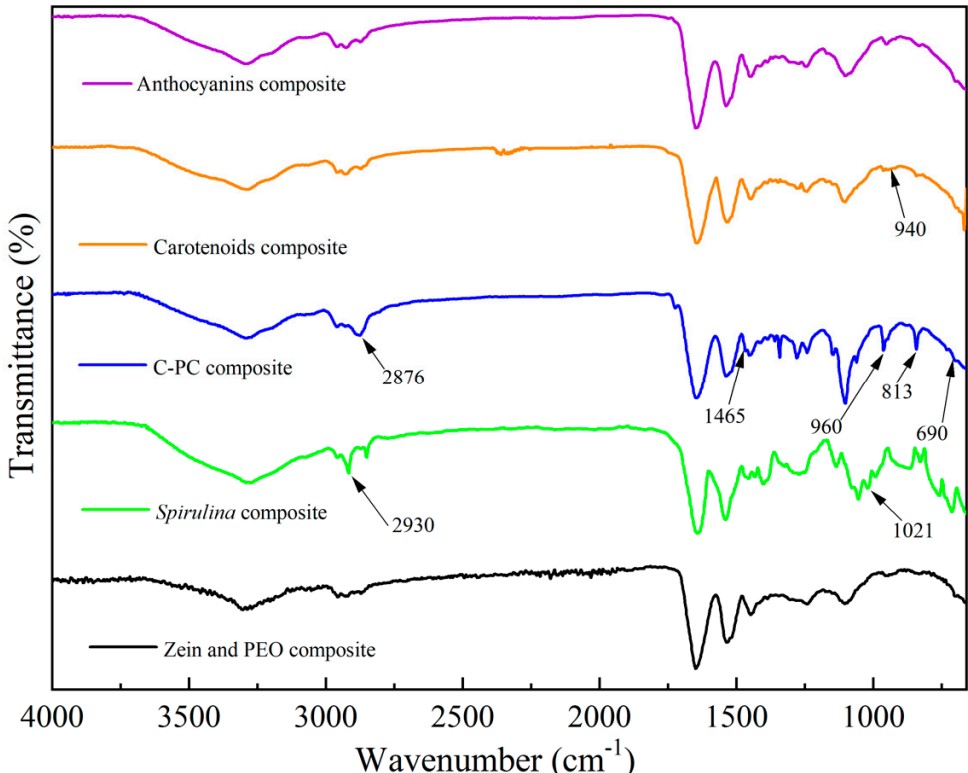

**Figure 6. The** ATR-FTIR spectrum of the composite Zein and PEO composite, *Spirulina* composite, C-PC composite, carotenoids composite and anthocyanins composite samples.

All modified samples present the characteristic bands of PEO and Zein, referring to sample 6. However, some of these bands appear in the same region of the spectrum of characteristic bands of the pigments, thus making it impossible to observe these bands, although some characteristic bands can still be identified.

## 4. Discussion

The results showed the strong influence of the composition of polymeric solutions on the rheological properties. Most solutions showed a shear-thinning behavior ($n < 1$), in which the viscosity of solutions decreased with the increase in shear rate. This behavior is widespread for polymeric solutions, which can be explained by the extensibility of polymer chains, together with other phenomena such as excluded volume and hydrodynamic interactions [28].

Polymeric solutions containing a higher amount of zein (10–12%) (samples 5, 6 and 7) showed lower viscosity values, indicating that the concentration of this protein did not directly affect the viscosity of polymeric blends. This property depends on the interaction between the zein and the PEO, which can occur through hydrophobic interactions, hydrogen bonding and van der Walls interactions. These interactions can increase the number of entanglements, resulting in a higher viscosity of blends than the solutions of individual polymers [17]. The results showed that the interactions with PEO were different depending on the biopolymer.

The presence of PEO also contributed to the increase in the shear thinning behavior of the blends. From Table 2, it is possible to observe that samples containing PEO showed lower values of flow behavior index (n). On the other hand, solutions without PEO showed n values higher than 0.9.

Balik and Argin [17] also verified an increase in shear thinning behavior for blends containing PEO and attributed this behavior to the high crystallinity of this polymer, which can contribute to the polymer chain alignment in the direction of flow. Moreover, blends without PEO showed relatively low and apparent viscosity of the same magnitude.

The addition of bioactive compounds affected the rheological behavior of the polymeric solution (Table 2). *Spirulina* was the bioactive compound that most influenced the rheological properties, leading to an increased apparent viscosity and shear thinning behavior. In addition to acting as a pigment, other works from the literature have already described the action of *Spirulina* as a structuring agent in biopolymer matrices [29]. Anthocyanin also increased the apparent viscosity, while C-phycocyanin and carotenoid promoted a decrease in this property but with a minor influence on the n values.

The oscillatory behavior of the solutions is shown in Figure 2. Most of the samples showed tan $\delta$ in all the frequency ranges. As tan $\delta$ corresponds to the relation G$'$/G$''$, values higher than 1 indicate the liquid-like characteristic of these blends, even for systems with very high viscosity (sample 2). A similar result was observed by Balik and Argin [17] for mixtures of PEO and pectin and by Rošic et al. [30] for alginate-PEO and chitosan-PEO blends. Nevertheless, samples containing *Spirulina* and anthocyanin showed a solid-like structure, with tan $\delta$ lower than 1. As described earlier, Spirulina can act as a structuring agent, which can be attributed to the carbohydrates and proteins presented in this material [29]. On the other hand, the effect of anthocyanin can be explained by its capacity to interact with proteins, such as zein, which can promote the formation of a protein network [31].

The results obtained regarding the FE-SEM demonstrate that the PEO and Zein [32] combination presents the most promising experimental development considering the conditions evaluated, in the present work, compared to the other variety of parameters, even collating with the mixture of PEO and gelatin (sample 7) or chitosan and PEO (sample 2). The results also showed that adding PEO to the combinations led to the formation of fibers.

Relating the formation of fibers with the rheological properties of solutions, it was possible to verify that the samples that did not form nanofibers (samples 1 and 4) were those that did not have a viscoelastic behavior. According to Rošic et al. [30], a liquid-like behavior is necessary to enable jet stabilization, but a minimum elasticity of solution is required to initialize jet formation. The blends that showed a viscoelastic behavior were the same ones with higher apparent viscosity values (samples 2, 6 and 7) and produced fibers with higher diameters. In addition, the higher values of tan $\delta$ (Figure 2A) led to greater diameters (Table 2). The exception was sample 5, which did not show an elastic character but formed fibers with high diameters. The same tendency between tan $\delta$ and fiber diameters was observed for samples containing the bioactive compounds, which can indicate that more elastic solutions tend to form thinner fibers. The results also showed no correlation between fiber diameter and solution viscosity, which agrees with the findings of Balik and Argin [17], who suggested that viscosity is not an indication of jet formation.

From the ATR-FTIR analysis, the gelatin polymer spectrum, sample 4, exhibits two characteristic bands at 1630 cm$^{-1}$ (C=O stretching vibration) and 1528 cm$^{-1}$ (amino group) [33]. The composite PEO and gelatin, sample 7, presents the two characteristic bands of gelatin and the bands at 2878 cm$^{-1}$ (C-H stretching vibrations), 1455 cm$^{-1}$ and 1340 cm$^{-1}$ refer to C-H bending vibrations; 967 cm$^{-1}$ (C-O stretching vibration) and 848 cm$^{-1}$ (antisymmetric stretching vibration of CH$_2$ group) are attributed to the PEO polymer [34–36]. Sample 1 (chitosan and gelatin) showed four characteristic bands that can be attributed to chitosan at 1530 cm$^{-1}$ (C-O-C antisymmetric stretching), 1157 cm$^{-1}$ (C-O stretching vibration), 1062 cm$^{-1}$ (C-O-C antisymmetric stretching) and 1019 cm$^{-1}$ (O-H bending vibration). The bands at 1400 and 1633 cm$^{-1}$ refer to the symmetric stretching of -COO groups and the C=O stretching vibration of gelatin [16]. Sample 2 (Chitosan and PEO), in addition to the characteristic bands of chitosan, presents bands at 2885 cm$^{-1}$ (CH$_2$ stretching), 1461 cm$^{-1}$ (asymmetric CH$_2$ bending), 1336 cm$^{-1}$ (symmetric CH$_2$ wagging and some C-C stretching),

965 cm$^{-1}$ (C-O stretching vibration) and 848 cm$^{-1}$ (antisymmetric stretching vibration of the CH$_2$ group) that are characteristic of PEO [34–36].

Figure 4b displays the spectra of samples 3, 5 and 6. The zein spectra exhibit bands at 3304 and 2959 cm$^{-1}$ that refer to the carbonyl groups. The intense band at 1639 cm$^{-1}$ can be attributed to the amide I; this band is related to the C=O stretching vibration and directly associated with the backbone conformation [37]. Finally, the bands at 1537 cm$^{-1}$ and 1231 cm$^{-1}$ were due to the angular deformation vibrations of the N-H bond (amide II) and the axial deformation vibrations of the C-N bond, respectively [38]. The FTIR spectra of samples 5 and 6 that present PEO and zein in their composition show the characteristic bands of zein. On the other hand, some typical bands of PEO appear in the same range as the zein bands, which makes their identification difficult. However, in sample 6, we can observe PEO characteristic bands at 1106 cm$^{-1}$, 955 cm$^{-1}$ and 840 cm$^{-1}$ that refer to C-O-C stretching: the CH$_2$-CH$_2$ rocking and C-O-C vibration modes and CH$_2$ rocking, respectively, Aziz et al. [39]. This difference between the spectra is because sample 6 (1%) has approximately 3.3 times more PEO than sample 5 (0.30%).

The addition of natural pigments changed the morphology and the band's presence in FE-SEM and ATR-FTIR analysis. The sample modified with the *Spirulina*, sample 8 (Figure 6), presents at 2930 cm$^{-1}$ the band of C-H aliphatic stretching vibration, and at 1021 cm$^{-1}$ there is the presence of the -C-O group from the cellulose structure. These bands are characteristic of *Spirulina* [40]. The sample spectrum with C-phycocyanin, sample 9, shows a band at 2876 cm$^{-1}$ that can be attributed to COO, CO and conjugated double bonds [41]. The bands at 1465 and 960 cm$^{-1}$ are associated with the CH$_2$ bending vibration and C-N tensile vibration (amine I), respectively [42]. The C–H bond appears at 813 and 690 cm$^{-1}$. In the spectrum of sample 10, which contains the carotenoid pigment, it was impossible to observe the characteristic bands of carotene at approximately 1650 cm$^{-1}$ and 1450 cm$^{-1}$, which are attributed to the –C=C bond and the -CH$_2$ bending vibration. In the same region appear the bands attributed to the vibration of the -C=O stretch of the amide groups of zein and the C-H stretch vibrations of PEO. However, the band at 940 cm$^{-1}$ can be assigned to the -CH of β-carotene [43,44]. In the spectra of sample 11 with jussara anthocyanins, only the bands belonging to ring vibrations in the region from 830 to 760 cm$^{-1}$ of the pigment can be observed [45]. Figure 7 was constructed to better visualize the diameter distribution of the fibers diameters.

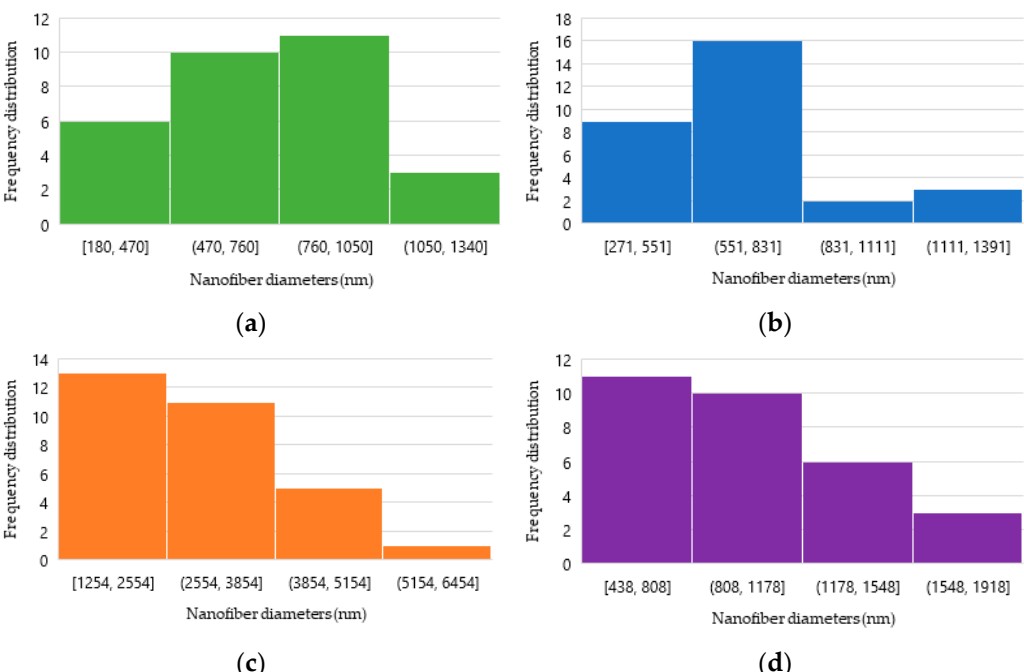

**Figure 7.** Histogram of average fiber diameter of samples 8 (**a**), 9 (**b**), 10 (**c**) and 11 (**d**).



Furthermore, from the data presented in Table 2, it is possible to observe that the addition of natural pigments did not significantly affect the diameter of the fibers formed when compared to the original solution (sample 6), except for the solution containing carotenoids (sample 10), which resulted in fibers with larger diameters. Studies on the production of zein nanofibers containing carotenoids have reported increased fiber thickness compared to solutions without the natural pigment. This fact may be related to the low solubility of carotenes in ethanol, which leads to a partial dispersion in the hydroalcoholic solution and the formation of carotene agglomeration, causing a thickening of the fibers formed [46,47].

## 5. Conclusions

The formation of polymeric blends was essential to achieve the desired morphology in the formation of nanofibers. The size of the diameters obtained from the fibers permits us to infer that solutions with elastic properties tend to form thinner fibers and adding PEO to the solutions ensures an increase in viscosity and shear thinning behavior, besides allowing a greater alignment of the formed fibers. Although the increase in viscosity with the addition of PEO has favored the formation of more homogeneous structures, it was not possible to relate this property to the diameter size of the fibers formed. The incorporation of natural pigments into the solutions caused variations in the rheology of the solutions, with the addition of *Spirulina* being the one that most altered the rheological properties. Nevertheless, the addition of natural pigments did not significantly affect the diameter of the fibers formed, with the exception of the formulation containing carotenoids that showed fibers with a higher diameter. Therefore, electrospinning is an excellent tool for obtaining nanostructures loaded with natural pigments and understanding how the rheological properties of the solution interfere with the formation of the nanostructures is essential in the process.

**Author Contributions:** Conceptualization, S.d.P.R., and A.R.C.B.; methodology, S.d.P.R., L.G.d.T., T.M.M. and A.R.C.B.; formal analysis, S.d.P.R., L.G.d.T., T.M.M. and A.R.C.B.; investigation, S.d.P.R., L.G.d.T., T.M.M., V.V.d.R., F.P.B. and A.R.C.B.; resources, V.V.d.R., F.P.B., E.L. and A.R.C.B.; writing—original draft preparation, S.d.P.R. and A.R.C.B.; writing—review and editing S.d.P.R., L.G.d.T., T.M.M., F.P.B., V.V.d.R., E.L. and A.R.C.B. All authors have read and agreed to the published version of the manuscript.

**Funding:** This work was supported by the "Fundação de Amparo à Pesquisa do Estado de São Paulo" (FAPESP) (process n° 2018/01550-8, 2018/13408-1, 2019/08975-1, 2019/26137-9, and 2020/06732-7) and 2020 GFI Competitive Grant Program.

**Data Availability Statement:** The data supporting reported results can be accessed via email for the corresponding author (anna.braga@unifesp.br).

**Acknowledgments:** A sincere thanks to Fazenda Tamanduá® for the organic powdered *Spirulina* donation.

**Conflicts of Interest:** The authors declare no conflict of interest.

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
