# Peer review of "Electrospinning Composites as Carriers of Natural Pigment: Screening of Polymeric Blends"

_processes, doi:10.3390/pr10122737_

Round 1
Reviewer 1 Report
Questions and suggestions:
1. No information about the materials of this study such as chitosan, Gelation and PEO etc. Please add all sample’s information into manuscript.
2. Why authors separate table 1 into 2 parts, see in page 2?
3. What is temperature on rheological test? And why author have investigated in the range of 0 and 300 s-1. Frequency and time sweep also should be investigated.
4. Page 5 line 154, auhor mentioned “The addition of Spirulina ………….began to behave like a solid-like structure (G’ > G”)” Why?
5. Histogram of average fiber diameter must be provided.
6. Conclusion is too generic. It is should be concise and showed scientific sentences.
Author Response
Responses to Technical Check Results
Santos, December 12th, 2022
Manuscript ID: processes-2097989
Title: Electrospinning composites as carriers of natural pigment: screening of polymeric blends
Dear Editor,
We would like to thank the reviewers for their valuable comments and suggestions for our manuscript. We have modified the manuscript according to the recommendations, and the detailed, point-by-point corrections are listed below. The modifications requested were extremely important to improve our work. All sentences/words listed below are highlighted in the revised version of the manuscript.
Reviewer 1:
Comment: No information about the materials of this study such as chitosan, Gelation and PEO etc. Please add all sample’s information into manuscript.
Response: The authors appreciate the opportunity to improve the manuscript. and the information was added to the text on Introduction.
Comment: Why authors separate table 1 into 2 parts, see in page 2?
Response: To be clear to the reader that the second part is regarding the natural pigments added.
Comment: What is temperature on rheological test? And why author have investigated in the range of 0 and 300 s-1. Frequency and time sweep also should be investigated.
Response: Rheological tests were performed at 25ºC. This information was previously added in the description of frequency sweep tests. In order to make the information clearer, the sentence was modified and presented separately.
Flow curves were investigated between 0 and 300 s-1 because this range is commonly used to characterize the flow behavior of fluids because it comprises typical processes of food industry, such as sedimentation, stirring and pipe flow (Steffe, 1996). In addition, the shear rate of the electrospinning process was within this range, as showed on Table 2. Frequency sweep was investigated, as shown in Figure 2.
Steffe, J. F. (1996). Rheological methods in food process engineering (2nd ed.). Michigan: Freeman Press.
Comment: Page 5 line 154, auhor mentioned “The addition of Spirulina ………….began to behave like a solid-like structure (G’ > G”)” Why?
Response: The information was added. The explanation of the effect of Spirulina and anthocyanins on the viscoelastic behavior of fluids, making them as solid-like structures, was added to the text on the discussion section.
Comment: Histogram of average fiber diameter must be provided.
Response: The histograms were added.
Comment: Conclusion is too generic. It is should be concise and showed scientific sentences.
Response: The conclusion was enhanced as requested.
Reviewer 2 Report
The presented article is devoted to the preparation of nanostructures containing natural pigments and the effect of these added pigments on the formation of nanostructures. The article is well written, but it would need a language proofreading and, above all, an improvement in the visualization of the measured results. Unification is needed in the units used as well as the reference format (see attached pdf).

Author Response
Responses to Technical Check Results
Santos, December 12th, 2022
Manuscript ID: processes-2097989
Title: Electrospinning composites as carriers of natural pigment: screening of polymeric blends
Dear Editor,
We would like to thank the reviewers for their valuable comments and suggestions for our manuscript. We have modified the manuscript according to the recommendations, and the detailed, point-by-point corrections are listed below. The modifications requested were extremely important to improve our work. All sentences/words listed below are highlighted in the revised version of the manuscript.
Reviewer 2: Comment: The presented article is devoted to the preparation of nanostructures containing natural pigments and the effect of these added pigments on the formation of nanostructures. The article is well written, but it would need a language proofreading and, above all, an improvement in the visualization of the measured results. Unification is needed in the units used as well as the reference format (see attached pdf).
Response: The authors appreciate the opportunity to improve the manuscript.
Comment: Units
Response: The units were added.
Comment: Consider expanding the introduction with a brief overview of all techniques for the preparation of nanostructures.
Response: The introduction was enhanced as requested.
Comment: Was the procedure from Reference 6?
Response: The information was added.
Comment: Clarify the composition of the mixture.
Response: The information was added.
Comment: Mass concentration percentage?
Response: The information was added.
Comment: What is the cation in the acetate buffer?
Response: The information was added.
Comment: References
Response: The information was added.
Comment: Consider enhancement of the layout and quality of the figure.
Response: The Figure was changed as requested.
Comment: Why the measurement of sample 6 starts at lower frequencies than others?
Response: Thank you for your comment. Figure 2 was modified so that all curves start from the same frequency.
Comment: References
Response: The information was added.
Comment: Consider marking the individual samples to the corresponding images.
Response: The Figure was changed as requested.
Comment: Consider marking the individual samples to the corresponding images.
Response: The Figure was changed as requested.
Comment: Consider adding a paragraph discussing the reasons for the influence of the added pigment on the formation of nanostructures (pigment structure, physico-chemical properties, etc.).
Response: The authors appreciate the opportunity to improve the manuscript, and the information was added to the text on Discussion.
Comment: Consider adding images of structures and tables of individual vibrations to better orient the reader.
Response: Table 3 was added as requested.
Comment: Unify the lower index
Response: The correction was made.
Comment: Characterize the mentioned Amid I and II.
Response: Information added as requested.
Comment: Unify the reference format
Response: The correction was made.
Round 2
Reviewer 1 Report
The authors answered and revised some questions and comments satisfactorily, but Figure 6 is very blurry. Besides, I did not see any revision in the part of the conclusion.
Author Response
Response to Reviewer 1 Comments
Santos, December 14th, 2022
Manuscript ID: processes-2097989
Title: Electrospinning composites as carriers of natural pigment: screening of polymeric blends
Dear Editor,
We would like to thank the reviewer again for the valuable comments and suggestions for our manuscript. We have modified the manuscript according to the recommendations, and the detailed, point-by-point corrections are listed below. All sentences/words listed below are highlighted in red in the revised version of the manuscript.
Reviewer 1:
Comment: Figure 6 is very blurry.
Response: As suggested by the reviewer, the quality of Figure 6 was improved.
Comment: I did not see any revision in the part of the conclusion.
Response: The conclusion was rewritten to become more concise and with more scientific information.